# Inhibition of *miR-143-3p* Restores Blood–Testis Barrier Function and Ameliorates Sertoli Cell Senescence

**DOI:** 10.3390/cells13040313

**Published:** 2024-02-08

**Authors:** Ziyan Xiao, Jinlian Liang, Rufei Huang, Derong Chen, Jiaxin Mei, Jingxian Deng, Zhaoyang Wang, Lu Li, Ziyi Li, Huan Xia, Yan Yang, Yadong Huang

**Affiliations:** 1Department of Cell Biology, Jinan University, Guangzhou 510632, China; xzy2021@stu2021.jnu.edu.cn (Z.X.); liangjinl01@gzhu.edu.cn (J.L.); sophie12@stu2022.jnu.edu.cn (R.H.); 3322nemo@stu2020.jnu.edu.cn (D.C.); 15617107825@163.com (J.M.); wzy1003@stu2021.jnu.edu.cn (Z.W.); lilu2022@stu2022.jnu.edu.cn (L.L.); lzy2023@stu2023.jnu.edu.cn (Z.L.); xiahuan@stu2019.jnu.edu.cn (H.X.); 2Department of Pharmacology, Jinan University, Guangzhou 510632, China; dengjingxian@stu2021.jnu.edu.cn; 3Guangdong Province Key Laboratory of Bioengineering Medicine, Guangzhou 510632, China

**Keywords:** *miR-143-3p*, sertoli cell, senescence, blood–testis barrier, UBE2E3

## Abstract

Due to the increasing trend of delayed childbirth, the age-related decline in male reproductive function has become a widely recognized issue. Sertoli cells (SCs) play a vital role in creating the necessary microenvironment for spermatogenesis in the testis. However, the mechanism underlying Sertoli cell aging is still unclear. In this study, senescent Sertoli cells showed a substantial upregulation of *miR-143-3p* expression. *miR-143-3p* was found to limit Sertoli cell proliferation, promote cellular senescence, and cause blood–testis barrier (BTB) dysfunction by targeting ubiquitin-conjugating enzyme E2 E3 (UBE2E3). Additionally, the TGF-β receptor inhibitor SB431542 showed potential in alleviating age-related BTB dysfunction, rescuing testicular atrophy, and reversing the reduction in germ cell numbers by negatively regulating *miR-143-3p*. These findings clarified the regulatory pathways underlying Sertoli cell senescence and suggested a promising therapeutic approach to restore BTB function, alleviate Sertoli cell senescence, and improve reproductive outcomes for individuals facing fertility challenges.

## 1. Introduction

In modern society, where the trend of delayed parenthood is rapidly gaining ground [1], age-related male reproductive decline is attracting increasing attention. Male fertility depends on the spermatogenesis process, which is carried out by the testis, the male reproductive organ in charge of creating sperm [2]. In testis, Sertoli cells (SCs) provide structural support and nutrients to spermatogonial stem cells [3,4] and transport nutrients to germ cells to maintain spermatogenesis [4,5,6,7]. The blood–testis barrier (BTB) formed by tight junctions (TJs) between adjacent SCs [8] divides the germ cells from the immune system, creating an immune-privileged site within the testis [9,10]. The main components of TJs are Claudins, Occludin, and JAM polymerized on the cell membrane, while ZOs serve as cytoplasm scaffolding proteins of TJs [11,12].

It has been proposed that SCs are among the cells most susceptible to age-associated dysfunction in the male reproductive system [13]. Age-associated decrease in the number of SCs and their dysfunction might frequently be the reason for spermatogenic failure [14,15]. Aging SCs display the reduction in quantity and the disturbance of metabolism, which can weaken their nutritional support to germ cells [14]. Additionally, the TJs between aging SCs are reduced and frequently replaced by focal contact points [16], which may suggest a damaged BTB. A broken BTB may lead to the dysfunction of spermatogenesis in varicocele patients [17]. It can also result in the death of germ cells and cause non-obstructive azoospermia (NOA) [18]. Therefore, delaying the aging of SCs and maintaining BTB functions is of great significance to male reproduction.

In the last several years, miRNAs have emerged as important participants in a number of cellular processes, including cell senescence and tissue homeostasis [19,20,21]. miRNAs in SCs are participating in the control of proliferation, maturation, apoptosis, and BTB formation [22]. DICER is a key enzyme for miRNA synthesis, and knocking out DICER always leads to disordered cell proliferation, increased apoptosis, defects in sperm morphology, and disruption of spermatogenesis [23]. Previous studies have shown that *miR-638* could promote apoptosis in porcine SCs by affecting PI3K (phosphoinositide 3 kinases) and AKT (Protein kinase B, PKB) required for the cell cycle [24]. *miR-202-3p* was increased in SCs of SCOS (Sertoli cell only syndrome) patients, and its overexpression inhibited SC proliferation and induced apoptosis [25]. In mouse SCs, miR-130 directly inhibited AR (androgen receptor) expression and resulted in defective spermatogenesis [26]. Similarly, it has been reported that the absence of DICER disrupts the formation of connections between cells in the seminiferous epithelium, thereby affecting the BTB [27]. The reasons may include the regulation of TJ proteins by multiple miRNAs [28]. For example, the *miR-181c/d*-Pafah1b1 axis disrupted the integrity of the BTB by disorganizing the localization of TJ proteins ZO-1 and Occludin [29]. And the TGF-β3-mediated BTB impairment could be rescued by *miR-142-3p* through the targeted knockdown of the Lgl2 (lethal giant larvae 2) [30]. Additionally, the overexpression of *miR-382-3p* in pubertal mice SCs resulting in a noteworthy decrease in Claudin-11, interrupted the functional maturation of SCs, thereby resulting in a defective BTB formation, even leading to infertility [31]. However, the miRNA regulatory mechanisms controlling SC aging and function remain largely unexplored.

In this work, we observed a substantial upregulation in the expression of *miR-143-3p* in aging SCs. Dysregulation of *miR-143-3p* leads to BTB impairment, resulting in disordered spermatogenesis and male infertility.

## 2. Materials and Methods

### 2.1. Ex Vivo Culture of Mouse SCs

SCs were isolated as we described previously from 3-week-old male Kunming mice [32,33]. In brief, the testes were aseptically isolated from mice, and washed 3 times in phosphate-buffered saline (PBS). Subsequently, the decapsulated testes were digested in Dulbecco’s Modified Eagle Medium (DMEM) (C11995500BT, Gibco, Grand Island, NY, USA) containing 500 μg/mL collagenase IV (C-5138, BioSharp, Beijing, China) at 37 °C by shaking in a water bath for 5–7 min. DMEM medium was used to terminate the digestion. After unit gravity sedimentation for 2 min, the Leydig cells enriched with supernatant were discarded. Then, the remaining tissues were digested in 0.25% trypsin (25200072, Gibco, Grand Island, NY, USA) at 37 °C for 10 min. The DMEM with 10% fetal bovine serum (FBS) (FSD500, ExCell Bio, Shanghai, China) was added to terminate the digestion. For separation and culture of SCs, the cell suspension was filtered through a 100-mesh filter screen, then the resultant filtrate was centrifuged at 1200× *g* rpm for 10 min to separate cells. SCs were cultured in DMEM with 10% FBS at 37 °C with 5% CO_2_; 24 h later, after the cells had been PBS-washed twice, the fresh medium was added to continue the cell culture.

### 2.2. Animals

C57BL/6J mice were bought from the Animal Center of Guangdong Medical Laboratory, China. For in vitro studies, primary SCs were isolated from KM mice, whereas C57BL/6J mice were employed for in vivo studies. In addition to having unlimited access to water and a normocaloric diet (NCD), mice were kept in rooms with 12 h cycles of light and dark. All animal studies were carried out in compliance with the National Institutes of Health’s standards for the care and use of animals and authorized by Jinan University’s Institutional Animal Care and Use Committee (Approval No. IACUC-20210630-04).

### 2.3. Cell Transfection

Cell transfection and co-transfection were conducted when the cell density achieved 60–70%. *Ube2e3*-siRNA, NC-siRNA, *miR-143-4p* mimic/inhibitor, and NC mimic/inhibitor were synthesized by RiboBio, Guangzhou, China. Table 1 contains a list of the sequences. The transfection concentration of all above was 30 nM. Cell transfection was performed by Lipofectamine™ 3000 (L3000015, Thermo Fisher Scientific, Waltham, MA, USA) and RNAiMAX (13778150, Thermo Fisher Scientific, Waltham, MA, USA).

### 2.4. Quantitative Real-Time PCR (qRT-PCR)

In accordance with the manufacturer’s instructions, the TRIzol reagent (15596026, Invitrogen, Carlsbad, CA, USA) was used to extract the total RNA from tissues and cells. The RNA was converted into cDNA using a reverse transcription kit (RR036A, Takara, Tokyo, Japan). The qRT-PCR with ChamQ SYBR qPCR Master Mix (Q311-02, Vazyme, Nanjing, China) was implemented on CFX96 Real-Time PCR Detection System (Bio-Rad, Hercules, CA, USA). The qPCR primers were designed according to the CDS sequences acquired on NCBI and created using the SnapGene (Version 3.2.1, GSL Biotech, Chicago, IL, USA) design software. Table 2 contains a list of primers used in qRT-PCR.

### 2.5. Western Blot Assay

Testicular tissues and cells’ proteins were isolated using RIPA lysis buffer (FD008, Fudebio, Hangzhou, China) with PhosSTOP™ (4906845001, Roche, Basel, Switzerland) and 0.1 mM PMSF (FD0100, Fudebio, Hangzhou, China). The BCA Protein Assay Kit (23225, Thermo Fisher Scientific, Waltham, MA, USA) was used to quantify the protein content. Then, 30 ug protein was separated by SDS-PAGE and transferred to the PVDF membrane (10600021, Cytiva, Marlborough, MA, USA). The membrane was blocked with 5% skim milk for 1 h, then incubated with primary antibody overnight and secondary antibody for 1 h. The primary antibodies are listed in Table 3. The membrane was visualized with ECL Western Blotting Substrate (K12045D50, Advansta, San Jose, CA, USA) and the bands were quantified by Image J software (1.48v, National Institutes of Health, Bethesda, MD, USA).

### 2.6. Treatment of SCs with SB431542

Primary SCs were passaged to the fourth generation (P4) to simulate the cellular aging process in vitro. SCs at P4 were planted in 6-well (3 × 10^5^ cells per well) or 24-well plates (6 × 10^4^ cells per well). When the cell density reached 70%, cells received treatment with 2, 5, and 10 μM SB431542 [34] (SB431542, a TGF-β receptor inhibitor) (S1067, Selleck, Houston, TX, USA) for 24 h or 48 h, the mRNA or protein level of interest was detected.

### 2.7. Transepithelial Electrical Resistance (TER) Measurements

SCs were plated on cell culture inserts (3470, Corning Incorporated, Corning, NY, USA) in 24-well plates (10 × 10^4^ cells per insert), and then transfected with siRNA or miRNA mimic for 24 h. The integrity of the SC barrier could be detected by measuring the TER value [35] with the Millicell ERS system (Millipore, Bedford, MA, USA). Every sample’s TER values were computed in TER_sample_ (Ω · cm^2^) = (R_sample_ − R_blank_) (Ω) × effective membrane area (cm^2^).

### 2.8. Luciferase Reporter Assay

The *miR-143-3p* mimic, wide-type (WT), and mutant-type (MUT) *Ube2e3* 3′-UTR sequences were synthesized. The psiCHECK-2 Vector was digested by restriction enzymes, and the synthetic WT and MUT fragments were inserted into the plasmid. Then, the recombinant reporter plasmids were co-transferred to 293T cells with *miR-143-3p* mimic or NC mimic. Following 48 h of transfection, dual glo Luciferase Assay System kit (E1910, Promega, Madison, WI, USA) was used to examine the cells. Firefly luciferase (F-luc) was utilized to normalize Renilla luciferase (R-luc) activity to assess the efficacy of reporter translation.

### 2.9. β-Gal Staining

After fixing the cells for 15 min with 4% paraformaldehyde, they were treated for 15 min at 37 °C with 1 μmol/L SPIDER-β-gal (SG02, Dojindo, Kumamoto, Japan). DAPI staining (AR1177, Boster Bio, Shanghai, China) was applied to the nuclei for 10 min. Following the wash with PBS, cells were observed using the LSM900 confocal microscope (Zeiss, Oberkochen, Germany).

### 2.10. Oil Red O Staining

As per the instructions of Oil Red O kit (G1262, Solarbio, Beijing, China). The SCs were fixed with fixative for 20–30 min and washed. Then, the SCs were dyed with ready-to-use staining for 10–20 min, followed by cleaning with 60% isopropanol to clear the background. The nuclei were stained with Hematoxylin (G1140, Solarbio, Beijing, China). After washing with distilled water, cells were observed using an Eclipse upright microscope (Nikon, Tokyo, Japan).

### 2.11. EdU Staining

As per the instructions of kFluor488 Click-iT EdU (KGA331, KeyGEN, Suzhou, China), a 20 μM EdU working solution was prepared. A working solution in equivalent volume was added, which contained 10 μM EdU to a 24-well plate, incubating for 24 h. Then, the cells underwent 4% PFA fixation and 0.5% Triton X-100 (T8200, Solarbio, Beijing, China) permeabilization, followed by washing with 3% BSA (9048-46-8, Aladdin, Wuhan, China). Then, the Click reaction solution for a 30 min incubation in the dark was added. DAPI staining (AR1177, Boster Bio, Shanghai, China) was applied to the nuclei. Cells were observed under LSM900 confocal microscope.

### 2.12. Biotin-NHS Ester Labeling Assay

Activated biotin can enter into the seminiferous tubules and label intratubular protein molecules through the damaged BTB. Here, Biotin-NHS ester was used to detect the integrity of BTB. Two weeks after the administration, 10 mg/mL sulfo-NHS-LC-Biotin (21335, Thermo Fisher Scientific, Waltham, MA, USA) in PBS was injected into the testes. Thirty minutes later, the mice were euthanized. Testes were sectioned at 6 μm after being embedded in O. C. T Compound (62534, Sakura Finetek, Tokyo, Japan). After five minutes of acetone fixation, frozen sections were treated with Alexa Fluor™ Streptavidin. (S11226, Thermo Fisher Scientific, Waltham, MA, USA). DAPI staining was applied to the nuclei. Fluorescence was observed with a fluorescence microscope.

### 2.13. Testicular Injection of miR-143-3p

The *miR-143-3p* overexpressing adeno-associated virus (AAV-143) (HH20210710GYQ-AAV01, HanBio, Shanghai, China) or the empty AAV vector (AAV-C) injections were carried out in male C57BL/6J. At 5 weeks of age, mice received a dose of AAV injection (25 μL in each testis), and at 7 weeks, they were killed. In total, 10 mice per AAV-143 group, and 10 mice per control group were studied.

### 2.14. Treatment of Mice with SB431542

SB431542 administration in vivo was carried out in 8-month-old male C57BL/6J. The mice were split into two groups to be intraperitoneally injected with 100 μL SB431542 (50 μM) (S1067, Selleck, Houston, TX, USA) or 100 μL PBS once a week for a period of 3 months (*n* = 10 per group).

### 2.15. Statistical Analysis

Data were presented as mean ± standard deviation (SD) of at least three independent experiments. One-way ANOVA or *t*-test function in GraphPad Prism 8 was used to assess mean differences. Differences in means were considered statistically significant at * *p* < 0.05; ** *p* < 0.01; *** *p* < 0.001; **** *p* < 0.0001.

## 3. Results

### 3.1. Role of miR-143-3p Overexpression in SC Senescence and BTB Dysfunction

During passages of primary SCs, we observed a rise in the number of β-gal-positive cells, indicating the aging of them (Figure 1A). Concurrent with this, an interesting observation was that the expression of *miR-143-3p* gradually rose in correlation with the aging of SCs (Figure 1B). Conversely, the levels of tight junction protein ZO-1 significantly dropped during cellular aging (Figure 1C). Interestingly, we also found a rise of *miR-143-3p* in the testes of 8-month-old mice (old) compared to 3-month-old mice (young) (Figure 1D). This finding suggests that the upregulation of *miR-143-3p* could be linked to age-related SC dysfunction.

In order to explore the role of *miR-143-3p* in regulating SC senescence, SCs were treated with the *miR-143-3p* mimic or the NC mimic. β-gal staining revealed that compared to control cells, transfection with the *miR-143-3p* mimic markedly raised the number of senescent cells (Figure 1E) and reduced the percentage of EdU-positive cells (Figure 1F). Next, we analyzed the expression of distinct senescence-associated secretory phenotype (SASP) factors, which is characterized by the activation of *Il-1β*, *Il-6*, *Il-10*, *Cxcl-1*, *Cxcl-2*, and *Tnf-α*. The results showed that compared to the NC group, the *miR-143-3p* mimic significantly up-regulated the expression of *Il-1β*, *Il-6*, *Il-10*, *Cxcl-1*, *Cxcl-2*, and *Tnf-α* (Figure 1G). One of the biological functions of SC was to phagocytize apoptotic germ cells and degrade them to form lipids [36] (Figure 1H). Therefore, the phagocytosis of SCs could be reflected by the content of intracellular lipid droplets. Following the treatment of the *miR-143-3p* mimic, in Oil Red O staining, we observed a reduction in lipid droplets in SCs (Figure 1I), indicating *miR-143-3p* weakens phagocytosis of apoptotic germ cells by SCs. To assess the permeability of SCs monolayers, the Transepithelial Electrical Resistance (TER) across these layers was measured (Figure 1J). The *miR-143-3p* mimic expressively reduced the TER value, suggesting disruption of the integrity of the SC barrier (Figure 1K). Furthermore, Western blot analysis confirmed that TJ proteins ZO-1 and Claudin-11 decreased following *miR-143-3p* mimic treatment (Figure 1L).

In conclusion, these results indicated that *miR-143-3p* promoted SC senescence, which subsequently affects their biological functions, including the weakening of their phagocytic ability and disruption of the tight junction integrity of the BTB.

### 3.2. miR-143-3p Promoted the Senescence of SCs by Targets UBE2E3

To confirm *miR-143-3p*’s regulatory function in SC senescence, SCs were further treated with a *miR-143-3p* inhibitor. As opposed to the results obtained with the *miR-143-3p* mimic, transfection with the inhibitor reduced the number of β-gal-positive cells (Figure 2A) and raised the proportion of EdU-positive cells (Figure 2B). Furthermore, it reduced the mRNA levels of *Il-β*, *Il-10*, *Cxcl-1*, and *Tnf-α* (Figure 2C). In addition, phagocytosis of apoptotic germ cells by SCs was further increased after *miR-143-3p* inhibitor treatment, as shown by increased Oil Red O staining droplets in SCs (Figure 2D). Interestingly, the suppression of *miR-143-3p* improved the integrity of the SC barrier as indicated by a rise in TER value (Figure 2E) and the increased levels in ZO-1 and Claudin-11 (Figure 2F). These results demonstrated that the suppression of *miR-143-3p* rescued SC senescence and subsequently improved the functions of BTB.

To investigate the underlying mechanism of the impact, we analyzed the possible targets of *miR-143-3p* responsible for regulating the aging of SCs. And we identified 3′UTR of *Ube2e3* as maybe a target for *miR-143-3p* (Figure 2G). This finding was further confirmed through luciferase reporter assays, in which the *miR-143-3p* mimic markedly suppressed the luciferase activity of the wild-type reporter vector. On the contrary, when the 3’ UTR of *Ube2e3* was altered, suppression was relieved. (Figure 2H).

To further validate the direct targeting relationship between *miR-143-3p* and UBE2E3, we treated SCs with the *miR-143-3p* mimic and inhibitor, respectively. qRT-PCR analysis revealed that the mimic decreased the mRNA level of *Ube2e3*, whereas the inhibition resulted in a significant upregulation of *Ube2e3* (Figure 2I). These findings were further confirmed by the protein level (Figure 2J,K). Furthermore, the exogenous overexpression of UBE2E3 can be countered by the *miR-143-3p* mimic (Figure 2L). In summary, these findings demonstrated that *miR-143-3p* promoted SC senescence and subsequently affected the functions of BTB by targeting UBE2E3.

### 3.3. miR-143-3p Overexpression in Testis Induced SCs Dysfunction

To confirm *miR-143-3p*’s regulation in SC senescence and BTB function in vivo, we injected an adeno-associated virus carrying *miR-143-3p* (AAV-143) into the testes of 4-week-old male mice to achieve overexpression *miR-143-3p*. Meanwhile, we injected the control mice with AAV carrying mock sequences (AAV-C) (Figure 3A). Following two weeks, *miR-143-3p* expression in mice testicles was considerably higher in the AAV-143 group than in the AAV-C group (Figure 3B). And the overexpression of *miR-143-3p* that resulted in a downregulation in the protein levels of UBE2E3 was also seen in the testes (Figure 3C,E).

We then evaluated the impact of *miR-143-3p* overexpression on testis function. We found that the SASP factors, including *Il-β*, *Il-6*, *Il-10*, and *Tnf-α* were significantly increased in the testes following treatment with AAV-143 (Figure 3D). Additionally, the senescence-related proteins P16 and P38 were also markedly elevated (Figure 3E). Conversely, the TJ proteins ZO-1, Occludin, and Claudin-11 were notably downregulated (Figure 3F,G), and the presence of sulfo-NHS LC-biotin was observed in the seminiferous tubules in AAV-143 treatment, revealing that the overexpression of *miR-143-3p* disrupted the integrity of the BTB (Figure 3H). The results demonstrated that *miR-143-3p* overexpression in the testes induced SC dysfunction in vivo.

### 3.4. Loss of UBE2E3 Induce SCs Senescence 

To confirm the regulatory function of UBE2E3 in the SC senescence, we treated SCs with *Ube2e3* siRNAs. β-gal staining indicated a significant increase in the number of positive cells in the si-*Ube2e3* group compared to the si-NC group (Figure 4A,B). Additionally, EdU analysis revealed a substantial decrease in cell proliferation upon *Ube2e3* knockdown (Figure 4C,D). Western blotting further confirmed the upregulation of senescence-related proteins P53, P16, and P38 in response to *Ube2e3* knockdown (Figure 4E). Moreover, the inhibition of *Ube2e3* expression resulted in a remarkable rise in SASP expression (*Il-1β*, *Il-6, Il-10*, *Cxcl-1*, *Cxcl-2*, and *Tnf-α*), consistent with the results observed with the *miR-143-3p* mimic (Figure 4F). Collectively, these findings suggest that *Ube2e3* knockdown promoted SC senescence.

### 3.5. Loss of UBE2E3 Induced SC Dysfunction

To investigate the impact of UBE2E3 on the functions of SCs, oil Red O staining was used to assess the phagocytosis of SCs toward apoptotic germ cells. The results revealed that transfection with *Ube2e3* siRNAs resulted in reduced lipid droplets in SCs, indicating a weakened phagocytic capacity toward apoptotic germ cells (Figure 5A). Additionally, we examined the role of UBE2E3 in regulating the formation of tight junctions in the BTB. Our findings suggest that the knockdown of *Ube2e3* markedly suppressed the expression of ZO-1 and Claudin-11 (Figure 5B–D) and also markedly decreased the TER value (Figure 5E), indicating a destruction in the integrity of the SC barrier. These findings suggest that the inhibition of UBE2E3 downregulates TJ protein expression, thereby disrupting the permeability of the BTB.

### 3.6. SB431542 Attenuated Age-Related SC Dysfunction by Inhibiting miR-143-3p and Upregulating UBE2E3

Intriguingly, we found that SB431542, an inhibitor of the TGF-β receptor, could downregulate *miR-143-3p* expression in vitro at concentrations of 1 µM, 2 µM, 5 µM, and 10 µM (Figure 6A). Consequently, we proposed that SB431542 might potentially mitigate the age-related dysfunction of SCs and restore BTB function by suppressing *miR-143-3p*.

The 8-month-old male mice were treated with SB431542 for 3 months (Figure 6B). We found that SB431542 treatment led to the downregulation of *miR-143-3p* expression (Figure 6C) while upregulating UBE2E3 expression (Figure 6D,G). Immunofluorescence staining and WB analysis demonstrated that SB431542 therapy increased the expression of TJ proteins ZO-1, Occludin, and Claudin-11, suggesting enhanced BTB integrity (Figure 6E–G). Furthermore, SB431542 treatment effectively reduced the SASP factors *Il-1β* and *Tnf-α* expression (Figure 6H). And histological examination (H&E staining) revealed a significant rise in seminiferous epithelium thickness and seminiferous tubule diameter in the testes treated with SB431542 (Figure 6I). To assess testicular germ cell proliferation, PCNA immunostaining was performed, demonstrating a notable rise in the quantity of PCNA-positive cells within the seminiferous tubule following SB431542 treatment (Figure 6J).

These findings suggest that SB431542 treatment has the potential to alleviate age-related BTB dysfunction, rescuing testicular atrophy and reversing the reduction in germ cell numbers by suppressing *miR-143-3p* and upregulating UBE2E3.

## 4. Discussion

For male reproduction, SCs are integral contributors to the microenvironment propitious for spermatogenesis, offering not only structural support but also conferring immune privilege [37], and facilitating signal transduction [6,7,38]. Therefore, maintaining optimal SC number and function throughout life is the key to healthy testicular function, including sperm output and androgen production [39]. It has been suggested that SCs are among the cells most susceptible to age-related dysfunction in the male reproductive system [13]. During normal aging, it is accompanied by a reduction in the number of SCs, morphological changes, organelle aging, abnormal hormone secretion, and BTB dysfunction [40]. Single-cell analysis of testis aging found that the genes upregulated in aging Sertoli cells were enriched in inflammation-related genes [14]. In addition, the stem cell factor (SCF)/c-kit signaling pathway between SCs (senders) and spermatogonia (receivers) decreased during aging, which might lead to blocked the self-renewal and differentiation of spermatogonia [14,41,42,43]. Overall, the inflammation and metabolic dysregulation accompanied by SC senescence may affect spermatogenesis in older men.

Knocking out DICER, a key enzyme for miRNA synthesis, is one of the methods for SC ablation [44]. It implies that miRNA plays an important role in the physiology of SC, which has been confirmed in subsequent studies [22,45]. *miR-143-3p* was considered to be a miRNA related to cardiovascular disease, and it has been shown to be dramatically elevated in the myocardium after myocardial infarction (MI) in both humans and mice [46,47]. In cardiomyocytes, the overexpression of *miR-143-3p* suppressed the expression of Yap and Ctnnd1, attenuating cell proliferation and heart regeneration, which could be reversed by Melatonin [48]. Additionally, in aging adipose stem cells (ASCs), *miR-143-3p* downregulation was accompanied by weakened cell proliferation and migration [49]. Our study demonstrated that the overexpression of *miR-143-3p* in SCs could block cell proliferation and induce expression of SASP factors, accelerating age-related Sertoli aging and dysfunction. To understand the potential mechanism of how *miR-143-3p* mediates SC aging, we identified UBE2E3 as the direct target of *miR-143-3p* in SCs. The *miR-143-3p* directly bound to the 3′UTR of *Ube2e3* played a central role in ubiquitin chain assembly and had a major role in determining the outcome of ubiquitylation [50,51], making it closely related to protein homeostasis and cellular senescence [52,53,54,55]. Previous studies have shown that UBE2E3 expression in the mouse brain decreases with age, leading to age-related motor dysfunction [56]. UBE2E3 has also been shown to play a crucial part in the proliferation of retinal pigment epithelial (RPE) cells, and its consumption results in a loss of the proliferation marker Ki-67 [53]. UBE2E3 depletion is accompanied by a unique SASP, and the upregulation of tumor suppressors p53 and cell cycle arrest markers p21 CIP1/WAF1 and p16 INK4a [51]. Similarly, in this study, we found that a loss of UBE2E3 also resulted in the inhibition of cell growth and the induced expression of SASP factors, including *Il-β*, *Il-6*, *Il-10*, *Cxcl-1*, *Cxcl-2*, and *Tnf-α*. The increase in p38 with si-*Ube2e3* suggests that p38MAPK may induce SASP in SCs by activating nuclear factor kappa-B (NF-κB) [57]. Previous studies have shown that the E3 ligase MDM2 is a well-validated regulator of p53 [58] and drives cellular senescence [59]. And our study suggests that the activation and upregulation of p53 and p21 after UBE2E3 knockdown were observed in SCs. 

Aging is a complicated process that can be caused by a multitude of cell-intrinsic and extrinsic stressors, such as telomere shortening, DNA damage, oncogene activation, radiation exposure, chemotherapy, heat stress, and oxidative stress [60,61]. Previous studies have shown that *miR-143-3p* has been implicated in the autophagy of cardiac progenitor cells generated by oxidative stress [62]. And the upregulation of *miR-143-3p* was observed in gastric mucosal injury caused by cold stress [63]. According to more recent research, *miR-143-3p* was markedly up-regulated in the testicular heat stress model and might be a crucial regulatory element influencing spermatogenesis in hot stress [64]. Similarly, research on retinal epithelial cells has demonstrated that Nrf2, a vital transcription factor that withstands oxidative stress, may be bound to and regulated by UBE2E3 [65,66]. Intriguingly, an association has been found between post-traumatic stress disorder and UBE2E3 expression in neurons [66,67]. Given our findings that the *miR-143-3p*/UBE2E3 axis induced Sertoli cell senescence, we proposed that factors inducing premature testicular aging—such as chemotherapy, oxidative stress, and heat stress—might lead to aberrant expression of *miR-143-3p* or UBE2E3.

BTB is one of the tightest blood–tissue barriers in mammals [68], which contributes to building an immune-privileged organ [69], and it regulates the entry and exit of substances [70] and serves as a dynamic ultrastructure to segregate cellular events during the epithelial cycle of spermatogenesis [71]. BTB is composed of four different cell junctions (TJs, ectoplasmic specializations, desmosomes, and gap junctions) [72]. The TJs between adjacent SCs, due to the functions of gate and fence, are the most important components of the BTB [71]. In rats, BTB assembly is completed by postnatal day ~18 to 21, coinciding with the time that SCs cease to divide [68]. Additionally, SCs are instrumental in orchestrating various signaling pathways, including TGF-β and FSH-mediated signaling [38,73], which not only modulate BTB function but also assist in the periodic restructuring necessary for germ cell passage [74,75,76]. Hormonal influences, particularly testosterone and FSH, further dictate this dynamic, dictating the synchronized remodeling of the BTB and Sertoli cell functions [6,37]. Since SCs are the cornerstone of BTB architecture, understanding the impact of age-related changes is vital for discerning the pathophysiological mechanisms underlying testicular barrier integrity and, by extension, male reproductive health.

SCs were susceptible to age-dependent effects, the tight junctions between aging SCs are reduced and frequently replaced by focal contact points [16]. In rats, the TJs in the testis gradually disintegrate, accompanied by an abnormal ultrastructure and the decreased expression of the TJ proteins ZO-1, Occludin, and Claudin-11 with aging [77]. Catriona Paul also found that the integrity of BTB decreased with age, and the anchoring junction proteins between germ cells and SCs markedly decreased [78]. In our study, the overexpression of *miR-143-3p* can effectively downregulate the TJ proteins ZO-1, Occludin, and Claudin-11, and the knockdown of *Ube2e3* can also lead to the same consequence. Based on the above studies that *miR-143-3p* is a regulator of SC senescence by targeting of *Ube2e3*, we speculate that the reduction in TJ protein caused by the overexpression of *miR-143-3p* may be related to SC aging. This means that the *miR-143-3p*/UBE2E3 axis can disrupt BTB function by inducing SC senescence. However, the molecular mechanism by which UBE2E3 regulates TJ proteins has not been elucidated in our study. Therefore, ongoing research examining the molecular mechanism by which UBE2E3 regulates TJ proteins ZO-1, Occludin, and Claudin-11 will provide valuable insights into the intricate interplay between *miR-143-3p*, UBE2E3, and BTB function. More importantly, many studies have shown that *miR-143-3p* is an SMAD-dependent TGF-β transcriptional target [79,80]. Therefore, we examined SB431542, a specific inhibitor of the TGF-β receptor [81], to inhibit the expression of *miR-143-3p*. Interestingly, we observed that SB431542 can expressively inhibit *miR-143-3p* expression, thereby rescuing BTB dysfunction in aging testis and improving spermatogenesis arrest. This afforded a theoretical foundation and support for SB431542 as a promising therapeutic drug against age-related SC dysfunction and spermatogenic disorders.

Therefore, our findings suggest that *miR-143-3p* and UBE2E3 participate in regulating age-related SC dysfunction and provide possible therapeutic targets for the treatment of age-related male reproductive disorders. Continued research in this field will further advance our knowledge and pave the way for the development of effective interventions to address male reproductive disorders.

## 5. Conclusions

In this study, we identified *miR-143-3p* as a regulator of SC senescence and BTB dysfunction by targeting of *Ube2e3*. The *miR-143-3p* overexpression upregulated the expression of senescence-related proteins (p53, p16, and p38) and SASP factors, while downregulating TJ proteins including Claudin-11, ZO-1, and Occludin leading to SC senescence and BTB dysfunction. Additionally, our results suggest the potential of SB431542, a TGF-β receptor inhibitor, in alleviating age-related BTB dysfunction, rescuing testicular atrophy, and reversing the reduction in germ cell numbers through negative regulation of *miR-143-3p* (Figure 7).

## Figures and Tables

**Figure 1 cells-13-00313-f001:**
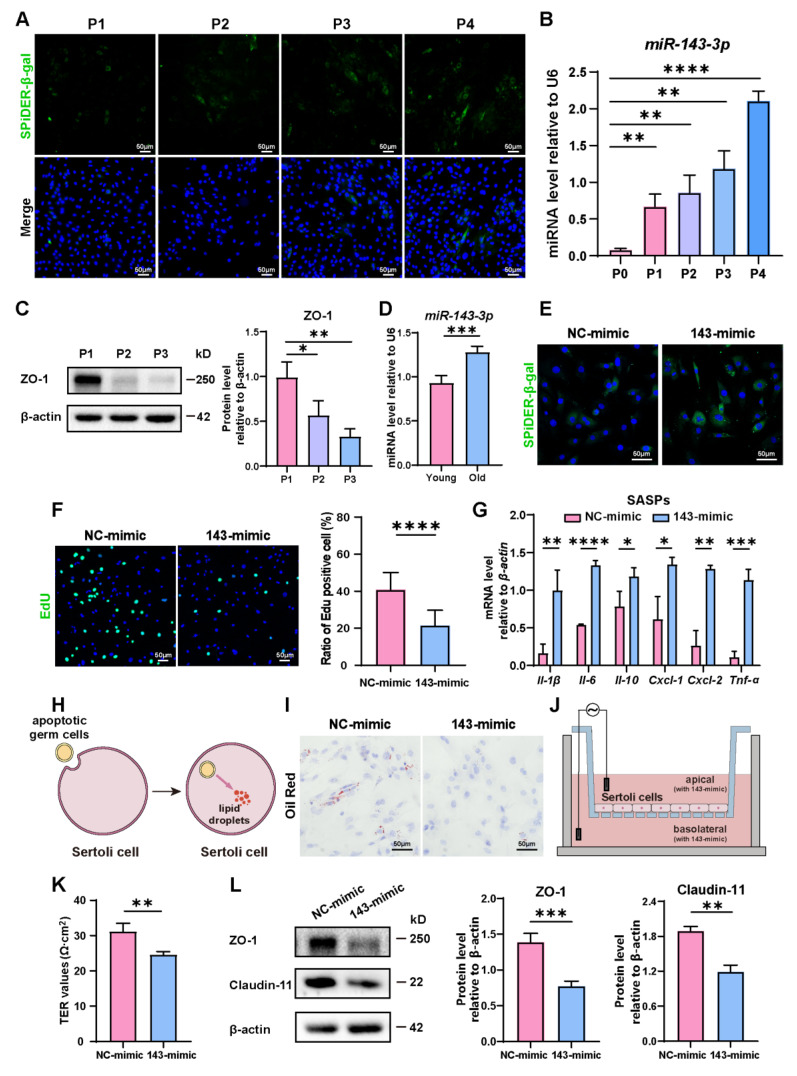
Role of *miR-143-3p* overexpression in SC senescence and BTB dysfunction: (**A**) Representative Senescence-associated β-galactosidase (SA-β-gal) staining (green) of SCs from different generations. DAPI as nucleus staining (blue). Scale bars, 50 µm. (**B**) *miR-143-3p* levels in different generations of SCs were detected by qRT-PCR. U6 was used as the internal reference gene. (**C**) Protein levels of tight junction (ZO-1) were detected by WB analysis, and β-actin was used as the internal reference protein. (**D**) qRT-PCR to analyze *miR-143-3p* levels in primary SCs isolated from testis of 3-month-old or 8-month-old mice. (**E**) SCs were stained with SA-β-gal (green) following a 48 h transfection with *miR-143-3p* mimic or NC mimic. DAPI as nucleus staining (blue). Scale bar, 50 µm. (**F**) Edu staining (green) detected the proliferation of SCs following a 48 h transfection with *miR-143-3p* mimic or NC mimic. DAPI as nucleus staining (blue). Scale bars, 50 µm. (**G**) qRT-PCR to analyze mRNA level of SASP factors in SCs with *miR-143-3p* mimics. (**H**,**I**) Oil Red O staining to perceive lipid droplets to reflect SC phagocytosis ability. Scale bars, 50 µm. (**J**,**K**) Transepithelial electrical resistance (TER) to evaluate the permeability of the SCs barrier. (**L**) WB analysis and quantitation of TJ proteins (ZO-1 and Claudin-11) in SCs treated with *miR-143-3p* mimic or NC mimic. Data are presented as the mean ± SD from at least three independent experiments. *p* < 0.05 was considered to be statistically significant. * *p* < 0.05; ** *p* < 0.01; *** *p* < 0.001; **** *p* < 0.0001.

**Figure 2 cells-13-00313-f002:**
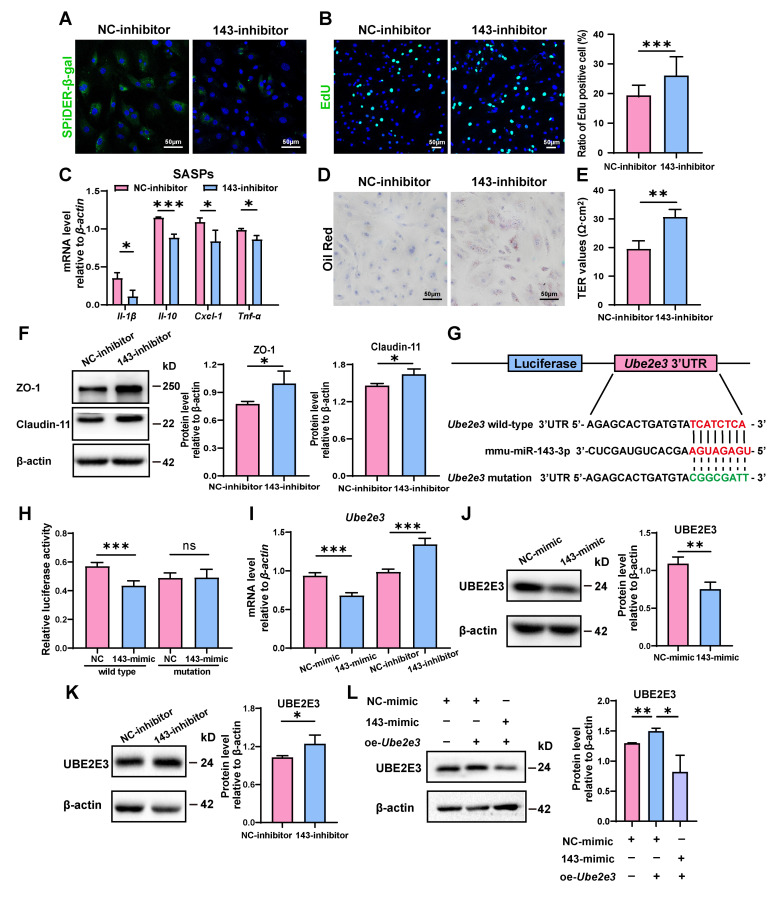
*miR-143-3p* promotes the senescence of SCs by targeting *Ube2e3*: (**A**) SCs were stained with SA-β-gal (green) after transfected *miR-143-3p* inhibitor or NC for 48 h. DAPI as nucleus staining (blue). Scale bar, 50 µm. (**B**) EdU (green) assays detected the proliferation of SCs treated with *miR-143-3p* inhibitor or NC inhibitor for 48 h. DAPI as nucleus staining (blue). Scale bars, 50 µm. (**C**) qRT-PCR analysis of SASP factors in SCs treated with *miR-143-3p* inhibitor or NC inhibitor for 24 h. (**D**) Oil Red O staining detected the phagocytosis of SCs treated with *miR-143-3p* inhibitor or NC inhibitor. Scale bars, 50 µm. (**E**) Permeability of the SC barrier was evaluated by TER. (**F**) WB analysis and quantification of ZO-1 and Claudin-11 in SCs treated with *miR-143-3p* inhibitor or NC inhibitor for 48 h. (**G**) *miR-143-3p* was predicted to attach to the 3′UTR region of *Ube2e3*. (**H**) The luciferase reporter vector and either the NC mimic or the *miR-143-3p* mimic were co-transfected into HEK293T cells. Relative R-luc activity was normalized to F-luc activity. (**I**) qRT-PCR to analyze *Ube2e3* levels in *miR-143-3p* mimic and inhibitor groups. (**J**,**K**) WB detection of UBE2E3 protein expression in *miR-143-3p* mimic and inhibitor groups. (**L**) WB detection of UBE2E3 expression after co-transfection of *miR-143-3p* mimic and UBE2E3 overexpression. Data are presented as the mean ± SD from at least three independent experiments. *p* < 0.05 was considered to be statistically significant. * *p* < 0.05; ** *p* < 0.01; *** *p* < 0.001 and ns = not significant.

**Figure 3 cells-13-00313-f003:**
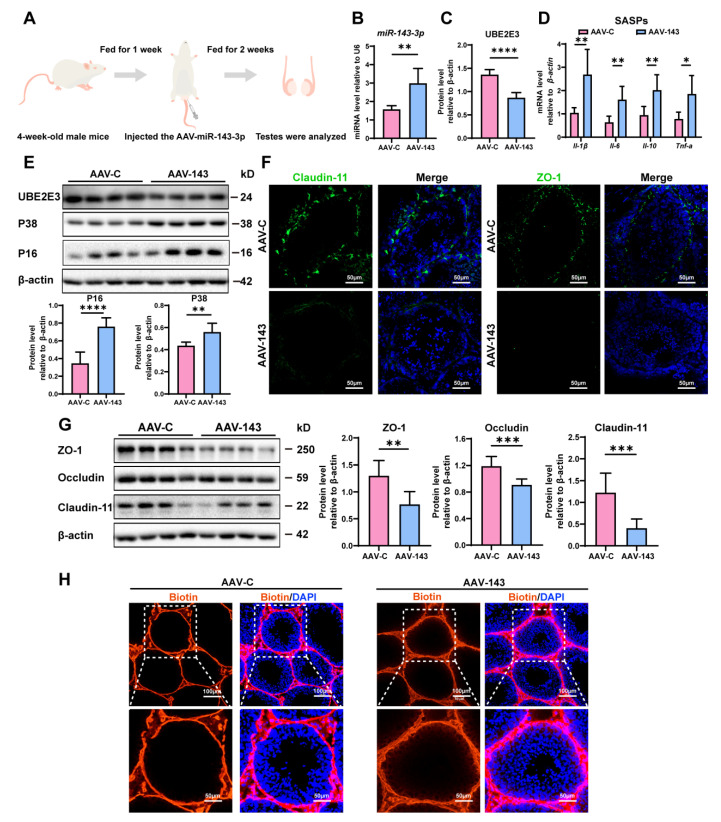
*miR-143-3p* overexpression in testis induced SCs dysfunction: (**A**) Schematic of AAV-143 injection in the mouse testis; 2 weeks after administration the mice were harvested for analysis. (**B**) qRT-PCR to analyze *miR-143-3p* expression in testes. U6 is the internal reference gene. (**C**) WB quantification of UBE2E3 in testes after AAV-143 treatment. (**D**) RT-qPCR to analyze SASP factors in mRNA level in testes with AAV-143 treatment. (**E**) WB analysis and quantification of UBE2E3 and senescence signaling pathway-related proteins. (**F**) Immunofluorescence (IF) staining of ZO-1 (green) and Claudin-11 (green). DAPI as nucleus staining (blue). Scale bars, 100 µm. (**G**) WB analysis and quantification of TJ proteins. (**H**) Biotin tracer (red) detects BTB permeability in mouse testes. DAPI as nucleus staining (blue). Scale bars, 50 µm and 100 µm. Data are presented as the mean ± SD of at least 6 animals. *p* < 0.05 was considered to be statistically significant. * *p* < 0.05; ** *p* < 0.01; *** *p* < 0.001; **** *p* < 0.0001.

**Figure 4 cells-13-00313-f004:**
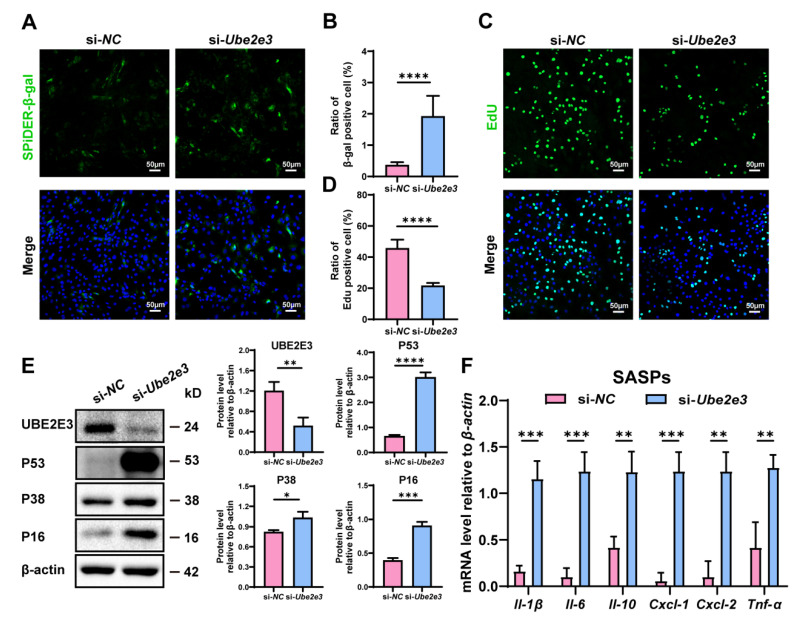
Loss of UBE2E3 induce SC senescence: (**A**,**B**) SCs were stained with SA-β-gal (green) after transfected si-*Ube2e3* for 48 h. DAPI as nucleus staining (blue). Scale bars, 50 µm. (**C**,**D**) Edu staining (green) to detect proliferation of SCs transfected with si-*Ube2e3* for 48 h. DAPI as nucleus staining (blue). Scale bars, 50 µm. (**E**) WB analysis and quantification of senescence-associated marker proteins of SCs after *Ube2e3* knockdown. (**F**) qRT-PCR analysis of mRNA level of the SASPs in SCs with si-*Ube2e3* treatment for 24 h. Data are presented as the mean ± SD of at least 3 independent experiments. *p* < 0.05 was considered to be statistically significant. * *p* < 0.05; ** *p* < 0.01; *** *p* < 0.001; **** *p* < 0.0001.

**Figure 5 cells-13-00313-f005:**
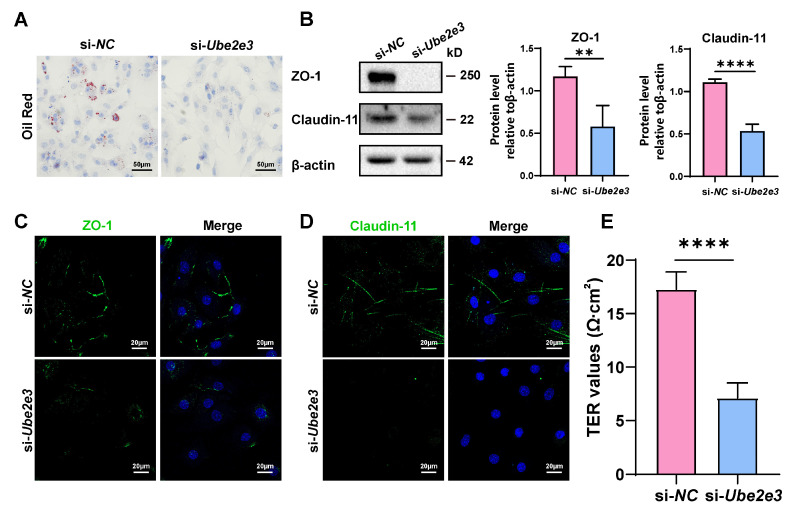
Loss of UBE2E3 induces SC dysfunction: (**A**) Oil Red O staining to detect SCs phagocytosis. Scale bars, 50 µm. (**B**) WB analysis and quantification of TJ proteins. (**C**,**D**) IF staining of TJ proteins ZO-1 (green) and Claudin-11 (green). DAPI as nucleus staining (blue). Scale bars, 20 µm. (**E**) TER to evaluate the permeability of the SC barrier. Data are presented as the mean ± SD of at least three independent experiments. *p* < 0.05 was considered to be statistically significant. ** *p* < 0.01; **** *p* < 0.0001.

**Figure 6 cells-13-00313-f006:**
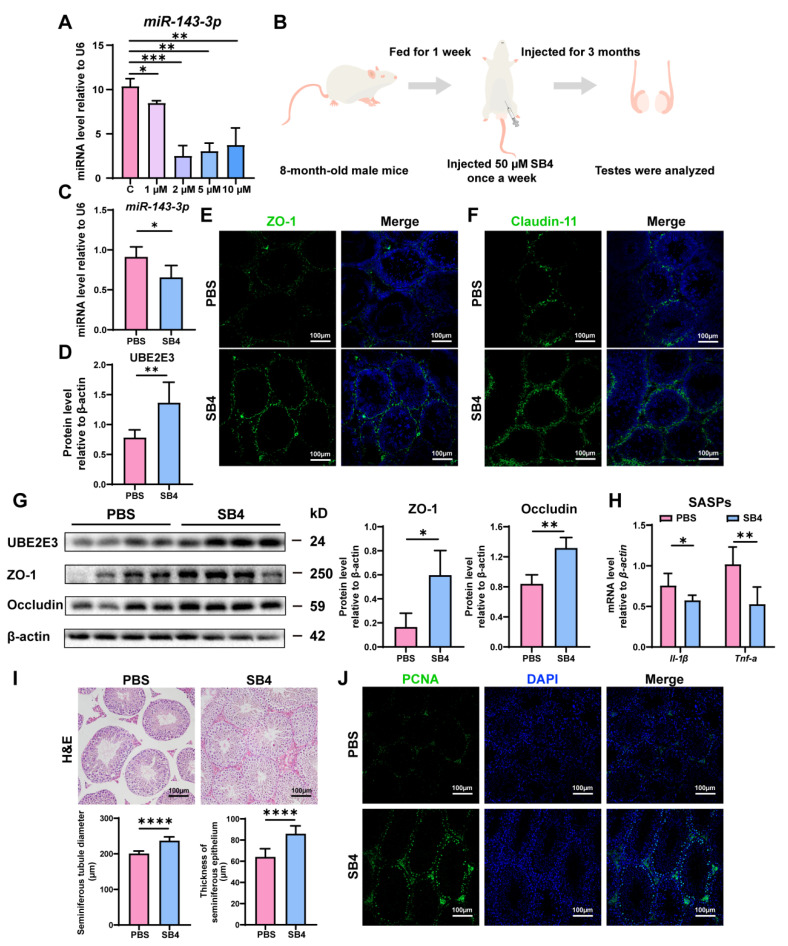
SB431542 attenuates testis aging and restores BTB function by targeting *miR-143-3p*: (**A**) The *miR-143-3p* miRNA levels in SCs treated with different concentrations SB431542 were detected by qRT-PCR. U6 is the internal reference. (**B**) Schematic diagram of intraperitoneal injection of SB431542. Eight-month-old male mice were injected intraperitoneally with SB431542 for 3 months. After the administration, the testis was collected to be analyzed. (**C**) qRT-PCR to analyze *miR-143-3p* levels in testes after SB431542 treatment. U6 is the internal reference gene. (**D**) WB quantification of UBE2E3 expression in testes after SB431542 treatment. (**E**,**F**) IF staining of ZO-1 (green) and Claudin-11 (green) in testes after SB431542 treatment. DAPI as nucleus staining (blue). Scale bars, 100 µm. (**G**) WB analysis of testicular TJ proteins after SB431542 treatment. (**H**) qRT-PCR to analyze mRNA level of the SASP factors in testes after SB431542 treatment. (**I**) H&E staining of the testis to analyze the seminiferous epithelium thickness and the seminiferous tubule diameter after SB431542 treatment. Scale bar, 100 µm. (**J**) IF staining of spermatogonia proliferation marker PCNA (green) after SB431542 treatment. DAPI as nucleus staining (blue). Scale bars, 100 µm. Data are presented as the mean ± SD of at least 6 animals. *p* < 0.05 was considered to be statistically significant. * *p* < 0.05; ** *p* < 0.01; *** *p* < 0.001; **** *p* < 0.0001.

**Figure 7 cells-13-00313-f007:**
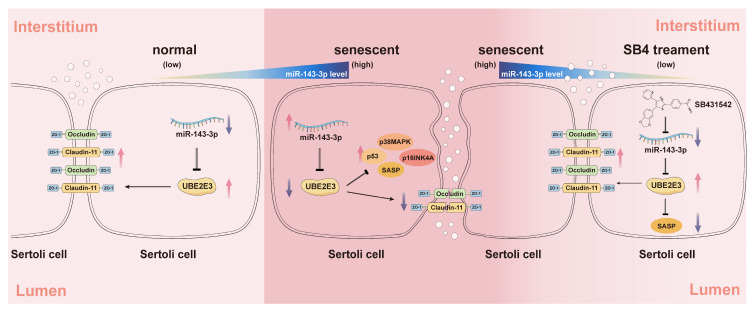
The schematic diagram illustrates the induction of SC senescence and subsequent regulation of the BTB by the *miR-143-3p*/UBE2E3 axis. *miR-143-3p* levels gradually increased in SCs during normal aging. In senescent SCs, the abnormally high expression of *miR-143-3p* activated the senescence signaling pathway p53, p38MAPK, p16INK4A, and SASPs by targeting and inhibiting the ubiquitin-conjugating enzyme UBE2E3. This results in the downregulation of tight junction protein to disrupt BTB integrity. The small molecule inhibitor SB431542 reversed this SC senescence and BTB dysfunction associated with abnormally *miR-143-3p* high expression. The red arrows stand for an increase in the expression, the purple arrows stand for a decrease. Black arrows stand for a positive effect of upstream factors on the protein it points to, instead, the arrows ending with a line stand for inhibition.

**Table 1 cells-13-00313-t001:** Sequences for miRNA-mimic/inhibitor and siRNA.

Name	Sequences (5′-3′)
*miR-143-3p*-mimic	UGAGAUGAAGCACUGUAGCUC
NC-mimic	UUUGUACUACACAAAAGUACUG
*miR-143-3p*-inhibitor	GAGCUACAGUGCUUCAUCUCA
NC-inhibitor	CAGUACUUUUGUGUAGUACAAA
si-*Ube2e3*	GCATAGCCACTCAGTATTT

**Table 2 cells-13-00313-t002:** Primers for qRT-PCR.

Name	Sense Primers (5′-3′)	Anti-Sense Primers
*β-actin*	GAGCGCAAGTACTCTGTGTG	AACGCAGCTCAGTAACAGTC
*Ube2e3*	TGCAACATCAACAGTCAGGGA	GAGTGGCTATGCTTCCGACC
*Il-1β*	GCCACCTTTTGACAGTGATGAG	GACAGCCCAGGTCAAAGGTT
*Il-6*	CACTTCACAAGTCGGAGGCT	CTGCAAGTGCATCATCGTTGT
*Il-10*	GGAGGGGTTCTTCCTTGGGA	TGAGCTGCTGCAGGAATGAT
*Cxcl-1*	TGCACCCAAACCGAAGTCAT	CTCCGTTACTTGGGGACACC
*Cxcl-2*	TCATAGCCACTCTCAAGGGC	TCAGGTACGATCCAGGCTTC
*Tnf-α*	ATGTCTCAGCCTCTTCTCATTC	GCTTGTCACTCGAATTTTGAGA

**Table 3 cells-13-00313-t003:** Antibody information.

Antibody	Source	Item Number	Working Dilution
p53	Proteintech (Wuhan, China)	10442	1:1000 (WB)
p38 MAPK	Cell Signaling Technology (Boston, MA, USA)	8690T	1:1000 (WB)
p16INK4a	Abcam (Cambridge, MA, USA)	Ab211542	1:1000 (WB)
β-actin	Fude BioTECH (Hangzhou, China)	FD0060	1:5000 (WB)
UBE2E3	Proteintech	15488	1:1000 (WB)
ZO-1	Abcam	Ab221547	1:100 (IF)–1:1000 (WB)
Occludin	Abcam	Ab216327	1:1000 (WB)
Claudin-11	Invitrogen (Carlsbad, CA, USA)	364500	1:100 (IF)–1:500 (WB)
PCNA	Abcam	Ab29	1:100 (IF)

## Data Availability

The data presented in this study are available on request from the corresponding authors.

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
