# Peer review of "Inhibition of miR-143-3p Restores Blood–Testis Barrier Function and Ameliorates Sertoli Cell Senescence"

_cells, 2024, doi:10.3390/cells13040313_

Round 1
Reviewer 1 Report
Comments and Suggestions for Authors
The authors showed the results of transepithelial electrical resistance (TER), and the values are approximately ~30 Ωꞏcm2, that seems to be too low to discuss. The author needs to describe whether these values are physiologically significant or not, and discuss the effect of miR-143-3p on tight junction proteins.
Comments on the Quality of English LanguageN/A
Author Response
Thank you for your insightful comments. We utilized an in vitro high-density Sertoli cell culture to mimic the blood-testis barrier (BTB), assessing its integrity through transepithelial electrical resistance (TER) measurements. Despite the lower TER values (~30 Ω·cm2) compared to typical standards (60-80 Ω·cm2) [1,2], these readings are considered reliable due to experimental variability introduced by factors such as cell density, experimental procedures, and equipment calibration.
Moreover, while TER values can provide an indication of BTB permeability, they do not fully reflect the physiological state of the BTB in vivo. We assessed physiological BTB permeability using the Biotin-NHS ester labeling assay in vivo, where biotin can not penetrate seminiferous tubules under normal physiological conditions. After AAV-miR-143-3p treatment, the presence of biotin in the tubules suggested a breakdown in BTB integrity. Further analysis showed miR-143-3p overexpression downregulates tight junction proteins ZO-1, Occludin, and Claudin-11, implying that miR-143-3p dysregulation can alter BTB function and permeability.
Additionally, we discuss the effect of miR-143-3p on tight junction proteins in revised manuscript. Please check line 492-502.
“ In rats, the TJs in testis gradually disintegrates accompanied by an abnormal ultrastructure and the decreased expression of the TJ proteins ZO-1, Occludin, and Claudin-11 with aging [77]. Catriona Paul also found that the integrity of BTB decreased with age, and the anchoring junction proteins between germ cells and SCs markedly decreased [78]. In our study, overexpression of miR-143-3p can effectively downregulate the TJ proteins ZO-1, occludin, and claudin-11, and the knockdown of Ube2e3 can also lead to the same consequence. Based on the above studies that miR-143-3p is a regulator of SC senescence by targeting of Ube2e3, we speculate that the reduction of TJ protein caused by the overexpression of miR-143-3p may be related to SC aging. It means that the miR-143-3p/UBE2E3 axis can disrupt BTB function by inducing SC senescence.”
1. Mruk, D.D.; Cheng, C.Y. An in vitro system to study Sertoli cell blood-testis barrier dynamics. Methods Mol Biol 2011, 763, 237-252, https://doi.org/10.1007/978-1-61779-191-8_16.
2. Wong, C.C.; Chung, S.S.; Grima, J.; Zhu, L.J.; Mruk, D.; Lee, W.M.; Cheng, C.Y. Changes in the expression of junctional and nonjunctional complex component genes when inter-sertoli tight junctions are formed in vitro. J Androl 2000, 21, 227-237.
77. Ma, Q.; You, X.; Zhu, K.; Zhao, X.; Yuan, D.; Wang, T.; Dun, Y.; Wu, J.; Ren, D.; Zhang, C.; et al. Changes in the tight junctions of the testis during aging: Role of the p38 MAPK/MMP9 pathway and autophagy in Sertoli cells. Exp Gerontol 2022, 161, 111729, https://doi.org/10.1016/j.exger.2022.111729.
78. Paul, C.; Robaire, B. Impaired function of the blood-testis barrier during aging is preceded by a decline in cell adhesion proteins and GTPases. PLoS One 2013, 8, e84354, https://doi.org/10.1371/journal.pone.0084354.
Reviewer 2 Report
Comments and Suggestions for Authors
Review: Inhibition of miR-143-3p restores blood-testis barrier function and ameliorates Sertoli cell senescence; Ziyan Xiao et al.
The manuscript describes the potential mechanism of Sertoli cell senescence during aging. It was observed that the miR-143-3p increased in Sertoli cells during aging in Kunming mice. This was associated with a loss of Sertoli cell proliferation, degradation of the Blood-Testis-Barrier (BTB), and atrophy of the seminiferous tubule with loss of germ cells. This aging was further shown to be associated with ubiquitin-conjugating enzyme E2 E3 (UBE2E3). Inhibition of the TGF-beta receptor with SB431542 was able to reverse the effects of aging. Molecular studies demonstrated this reversal by negatively regulating miR-143-3p. The studies provide an interesting new twist to the regulation of Sertoli cell function and sperm production.
This reviewer was particularly impressed with the multiple ways in which effects were demonstrated including histochemistry, PCR studies in different generations of Sertoli Cells, western blot analysis, and various transfection approaches. The approaches led to a most compelling story of the effects going on and rescue by TGF-beta receptor inhibition. Importantly the authors also demonstrated physiological effects of BTB disruption via transepithelial electrical resistance. The study was then completed with the presentation of a model of what was proposed to be happening within Sertoli cells to cause the observed responses.
The study did confine itself to senescence but a more general comment on other ways in which this could be occurring such as with heat stress, chemotherapy, and other stress responses would have been welcomed.
In general, this is a well-written manuscript that is very clear, the English is excellent, the Tables and figures are clear, experiments have well-reasoned hypotheses that were tested appropriately by the experimental designs. Thus, the conclusions are well supported by the evidence presented.
This is a novel idea and advances the knowledge of testis function during aging. It fits within the scope of the journal and should be of interest to the readers. I would give this article my highest degree of merit commendation.
Author Response
We are grateful for your comprehensive feedback. Your insights are greatly valued and have motivated us to delve deeper into our research.
In response to your suggestions, we have updated the manuscript to provide a more comprehensive discussion on the potential impacts of various stressors on Sertoli cell function and aging. This expanded discussion reflects on the ways in which factors such as heat stress, chemotherapy, and oxidative stress might intersect with the molecular mechanisms we have outlined, particularly focusing on the roles of miR-143-3p and UBE2E3. Please check line 460-474.
“ Aging is a complicated process that can be caused by a multitude of cell intrinsic and extrinsic stressors, such as telomere shortening, DNA damage, oncogene activation, radiation exposure, chemotherapy, heat stress and oxidative stress [60,61].Previous studies have shown that miR-143-3p has been implicated in the autophagy of cardiac progenitor cells generated by oxidative stress [62]. And the upregulation of miR-143-3p was observed in gastric mucosal injury caused by cold stress [63]. According to more recent research, miR-143-3p was markedly up-regulated in the testicular heat stress model and might be a crucial regulatory element influencing spermatogenesis in hot stress [64]. Similarly, research on retinal epithelial cells has demonstrated that Nrf2, a vital transcription factor that withstands oxidative stress, may be bound to and regulated by UBE2E3 [65,66]. Intriguingly, an association has been found between post-traumatic stress disorder and UBE2E3 expression in neurons [66,67]. Given our findings that the miR-143-3p/UBE2E3 axis induced Sertoli cell senescence, we proposed that factors inducing premature testicular aging—such as chemotherapy, oxidative stress, and heat stress—might lead to aberrant expression of miR-143-3p or UBE2E3.”
60. He, S.; Sharpless, N.E. Senescence in Health and Disease. Cell 2017, 169, 1000-1011, https://doi.org/10.1016/j.cell.2017.05.015.
61. Qin, S.; Schulte, B.A.; Wang, G.Y. Role of senescence induction in cancer treatment. World J Clin Oncol 2018, 9, 180-187, https://doi.org/10.5306/wjco.v9.i8.180.
62. Ma, W.; Ding, F.; Wang, X.; Huang, Q.; Zhang, L.; Bi, C.; Hua, B.; Yuan, Y.; Han, Z.; Jin, M.; et al. By Targeting Atg7 MicroRNA-143 Mediates Oxidative Stress-Induced Autophagy of c-Kit(+) Mouse Cardiac Progenitor Cells. EBioMedicine 2018, 32, 182-191, https://doi.org/10.1016/j.ebiom.2018.05.021.
63. Yin, Y.C.; Li, X.H.; Rao, X.; Li, Y.J.; Du, J. Involvement of microRNA/cystine/glutamate transporter in cold-stressed gastric mucosa injury. Front Pharmacol 2022, 13, 968098, https://doi.org/10.3389/fphar.2022.968098.
64. Gan, M.; Jing, Y.; Xie, Z.; Ma, J.; Chen, L.; Zhang, S.; Zhao, Y.; Niu, L.; Wang, Y.; Li, X.; et al. Potential Function of Testicular MicroRNAs in Heat-Stress-Induced Spermatogenesis Disorders. Int J Mol Sci 2023, 24, https://doi.org/10.3390/ijms24108809.
65. Plafker, K.S.; Nguyen, L.; Barneche, M.; Mirza, S.; Crawford, D.; Plafker, S.M. The ubiquitin-conjugating enzyme UbcM2 can regulate the stability and activity of the antioxidant transcription factor Nrf2. J Biol Chem 2010, 285, 23064-23074, https://doi.org/10.1074/jbc.M110.121913.
66. Huang, Z.; Zhang, D.; Chen, S.C.; Huang, D.; Mackey, D.; Chen, F.K.; McLenachan, S. Mitochondrial Dysfunction and Impaired Antioxidant Responses in Retinal Pigment Epithelial Cells Derived from a Patient with RCBTB1-Associated Retinopathy. Cells 2023, 12, https://doi.org/10.3390/cells12101358.
67. Wendt, F.R.; Pathak, G.A.; Levey, D.F.; Nuñez, Y.Z.; Overstreet, C.; Tyrrell, C.; Adhikari, K.; De Angelis, F.; Tylee, D.S.; Goswami, A.; et al. Sex-stratified gene-by-environment genome-wide interaction study of trauma, posttraumatic-stress, and suicidality. Neurobiol Stress 2021, 14, 100309, https://doi.org/10.1016/j.ynstr.2021.100309.
Reviewer 3 Report
Comments and Suggestions for Authors
Ziyan Xiao et al. in “Inhibition of miR-143-3p restores blood-testis barrier function and ameliorates Sertoli cell senescence”, summarized recent findings about the role of miR-143-3p as a regulator of SC senescence and BTB dysfunction by targeting of Ube2e. Moreove, they suggested that miR-143-3p and UBE2E3 played a role in regulating age-related SC dysfunction and provided potential therapeutic targets for the treatment of age-related male reproductive disorders.
The article paper is well written and very original. It is an in-depth and very articulated study.
The authors should better explain:
- the relationship between SCs and age-related dysfunction in male reproductive system;
- the relationship between SCs and maintaining BTB functions.
Comments on the Quality of English LanguageMinor editing of English language required
Author Response
Comments 1. The authors should better explain the relationship between SCs and age-related dysfunction in male reproductive system.
Response 1: We appreciate your valuable suggestions for improving the quality of our work. As your recommendation, we have added this content to the “ Discussion section”. Please check line 416-430.
“ For male reproduction, SCs are integral contributors to the microenvironment propitious for spermatogenesis, offering not only structural support but also conferring immune privilege [37], and facilitating signal transduction [6,7,38]. Therefore, maintaining optimal SCs number and function throughout life is the key to healthy testicular function, including sperm output and androgen production [39].It has been suggested that SCs are among the cells most susceptible to age-related dysfunction in the male reproductive system [13]. During normal aging, it is accompanied by the reduction in the number of SCs, the morphological changes, the organelle aging, the abnormal hormone secretion and the BTB dysfunction [40]. Single-cell analysis of testis aging found that the genes upregulated in aging Sertoli cells were enriched for inflammation-related Genes[14]. In addition, the stem cell factor (SCF)/c-kit signaling pathway between SCs (senders) and spermatogonia (receivers) decreased during aging, which might lead to blocked the self-renewal and differentiation of spermatogonia [14,41-43]. Overall, the inflammation and metabolic dysregulation accompanied with SC senescence may affect spermatogenesis in older men.”
6. Smith, L.B.; Walker, W.H. The regulation of spermatogenesis by androgens. Semin Cell Dev Biol 2014, 30, 2-13, https://doi.org/10.1016/j.semcdb.2014.02.012.
7. Gupta, A.; Vats, A.; Ghosal, A.; Mandal, K.; Sarkar, R.; Bhattacharya, I.; Das, S.; Pal, R.; Majumdar, S.S. Follicle-stimulating hormone-mediated decline in miR-92a-3p expression in pubertal mice Sertoli cells is crucial for germ cell differentiation and fertility. Cell Mol Life Sci 2022, 79, 136, https://doi.org/10.1007/s00018-022-04174-9.
13. Cummins, J.M.; Jequier, A.M.; Kan, R. Molecular biology of human male infertility: links with aging, mitochondrial genetics, and oxidative stress? Mol Reprod Dev 1994, 37, 345-362, https://doi.org/10.1002/mrd.1080370314.
14. Nie, X.; Munyoki, S.K.; Sukhwani, M.; Schmid, N.; Missel, A.; Emery, B.R.; DonorConnect; Stukenborg, J.B.; Mayerhofer, A.; Orwig, K.E.; et al. Single-cell analysis of human testis aging and correlation with elevated body mass index. Dev Cell 2022, 57, 1160-1176.e1165, https://doi.org/10.1016/j.devcel.2022.04.004.
37. Stanton, P.G. Regulation of the blood-testis barrier. Semin Cell Dev Biol 2016, 59, 166-173, https://doi.org/10.1016/j.semcdb.2016.06.018.
38. Ruthig, V.A.; Lamb, D.J. Updates in Sertoli Cell-Mediated Signaling During Spermatogenesis and Advances in Restoring Sertoli Cell Function. Front Endocrinol (Lausanne) 2022, 13, 897196, https://doi.org/10.3389/fendo.2022.897196.
39. O'Donnell, L.; Smith, L.B.; Rebourcet, D. Sertoli cells as key drivers of testis function. Semin Cell Dev Biol 2022, 121, 2-9, https://doi.org/10.1016/j.semcdb.2021.06.016.
40. Santiago, J.; Silva, J.V.; Alves, M.G.; Oliveira, P.F.; Fardilha, M. Testicular Aging: An Overview of Ultrastructural, Cellular, and Molecular Alterations. J Gerontol A Biol Sci Med Sci 2019, 74, 860-871, https://doi.org/10.1093/gerona/gly082.
41. Schrans-Stassen, B.H.; van de Kant, H.J.; de Rooij, D.G.; van Pelt, A.M. Differential expression of c-kit in mouse undifferentiated and differentiating type A spermatogonia. Endocrinology 1999, 140, 5894-5900, https://doi.org/10.1210/endo.140.12.7172.
42. Rossi, P.; Sette, C.; Dolci, S.; Geremia, R. Role of c-kit in mammalian spermatogenesis. J Endocrinol Invest 2000, 23, 609-615, https://doi.org/10.1007/bf03343784.
43. Mauduit, C.; Hamamah, S.; Benahmed, M. Stem cell factor/c-kit system in spermatogenesis. Hum Reprod Update 1999, 5, 535-545, https://doi.org/10.1093/humupd/5.5.535.
Comments 2. The authors should better explain the relationship between SCs and maintaining BTB functions.
Response 2: We appreciate your valuable suggestions for improving the quality of our work. As your recommendation, we have added this content to the " Discussion section". Please check line 475-490.
" BTB is one of the tightest blood-tissue barriers in mammals [68], which contribute to build a immune-privileged organ [69], regulate the entry and exit of substances [70] and serve as a dynamic ultrastructure to segregate cellular events during the epithelial cycle of spermatogenesis [71].BTB is composed of 4 different cell junctions (TJs, ectoplasmic specializations, desmosomes and gap junctions) [72]. The TJs between adjacent SCs, due to the functions of gate and fence, is the most important component of the BTB [71]. In rats, the BTB assembly completed by postnatal day ~ 18 to 21, coinciding with the time that SCs cease to divide [68]. Additionally, SCs are instrumental in orchestrating various signaling pathways, including TGF-β and FSH-mediated signaling [38,73], which not only modulate BTB function but also assist in the periodic restructuring necessary for germ cell passage [74-76]. Hormonal influences, particularly testosterone and FSH, further dictate this dynamic, dictating the synchronized remodeling of the BTB and Sertoli cell functions [6,37]. Since SCs are the cornerstone of BTB architecture, understanding the impact of age-related changes is vital for discerning the pathophysiological mechanisms underlying testicular barrier integrity and, by extension, male reproductive health."
6. Smith, L.B.; Walker, W.H. The regulation of spermatogenesis by androgens. Semin Cell Dev Biol 2014, 30, 2-13, https://doi.org/10.1016/j.semcdb.2014.02.012.
37. Stanton, P.G. Regulation of the blood-testis barrier. Semin Cell Dev Biol 2016, 59, 166-173, https://doi.org/10.1016/j.semcdb.2016.06.018.
38. Ruthig, V.A.; Lamb, D.J. Updates in Sertoli Cell-Mediated Signaling During Spermatogenesis and Advances in Restoring Sertoli Cell Function. Front Endocrinol (Lausanne) 2022, 13, 897196, https://doi.org/10.3389/fendo.2022.897196.
68. Cheng, C.Y.; Mruk, D.D. The blood-testis barrier and its implications for male contraception. Pharmacol Rev 2012, 64, 16-64, https://doi.org/10.1124/pr.110.002790.
69. Li, N.; Wang, T.; Han, D. Structural, cellular and molecular aspects of immune privilege in the testis. Front Immunol 2012, 3, 152, https://doi.org/10.3389/fimmu.2012.00152.
70. Pelletier, R.M. The blood-testis barrier: the junctional permeability, the proteins and the lipids. Prog Histochem Cytochem 2011, 46, 49-127, https://doi.org/10.1016/j.proghi.2011.05.001.
71. Mruk, D.D.; Cheng, C.Y. The Mammalian Blood-Testis Barrier: Its Biology and Regulation. Endocr Rev 2015, 36, 564-591, https://doi.org/10.1210/er.2014-1101.
72. Pelletier, R.M.; Byers, S.W. The blood-testis barrier and Sertoli cell junctions: structural considerations. Microsc Res Tech 1992, 20, 3-33, https://doi.org/10.1002/jemt.1070200104.
73. Ni, F.D.; Hao, S.L.; Yang, W.X. Multiple signaling pathways in Sertoli cells: recent findings in spermatogenesis. Cell Death Dis 2019, 10, 541, https://doi.org/10.1038/s41419-019-1782-z.
74. Lui, W.Y.; Lee, W.M.; Cheng, C.Y. TGF-betas: their role in testicular function and Sertoli cell tight junction dynamics. Int J Androl 2003, 26, 147-160, https://doi.org/10.1046/j.1365-2605.2003.00410.x.
75. Eto, K.; Shiotsuki, M.; Sakai, T.; Abe, S. Nociceptin is upregulated by FSH signaling in Sertoli cells in murine testes. Biochem Biophys Res Commun 2012, 421, 678-683, https://doi.org/10.1016/j.bbrc.2012.04.061.
76. Wang, Y.; Lui, W.Y. Opposite effects of interleukin-1alpha and transforming growth factor-beta2 induce stage-specific regulation of junctional adhesion molecule-B gene in Sertoli cells. Endocrinology 2009, 150, 2404-2412, https://doi.org/10.1210/en.2008-1239.